# The Role of Remote Sensing and Geospatial Analysis for Understanding COVID-19 Population Severity: A Systematic Review

**DOI:** 10.3390/ijerph20054298

**Published:** 2023-02-28

**Authors:** Butros M. Dahu, Khuder Alaboud, Avis Anya Nowbuth, Hunter M. Puckett, Grant J. Scott, Lincoln R. Sheets

**Affiliations:** 1Institute for Data Science and Informatics, University of Missouri, Columbia, MO 65211, USA; 2Department of Health Management and Informatics, University of Missouri, Columbia, MO 65211, USA; 3NextGen Biomedical Informatics Center, University of Missouri, Columbia, MO 65211, USA; 4Pan African Organization for Health Education and Research (POHER), Manchester, MO 63011, USA; 5Department of Electrical Engineering and Computer Science, University of Missouri, Columbia, MO 65211, USA

**Keywords:** remote sensing, geospatial, satellite imaging, COVID-19

## Abstract

Remote sensing (RS), satellite imaging (SI), and geospatial analysis have established themselves as extremely useful and very diverse domains for research associated with space, spatio-temporal components, and geography. We evaluated in this review the existing evidence on the application of those geospatial techniques, tools, and methods in the coronavirus pandemic. We reviewed and retrieved nine research studies that directly used geospatial techniques, remote sensing, or satellite imaging as part of their research analysis. Articles included studies from Europe, Somalia, the USA, Indonesia, Iran, Ecuador, China, and India. Two papers used only satellite imaging data, three papers used remote sensing, three papers used a combination of both satellite imaging and remote sensing. One paper mentioned the use of spatiotemporal data. Many studies used reports from healthcare facilities and geospatial agencies to collect the type of data. The aim of this review was to show the use of remote sensing, satellite imaging, and geospatial data in defining features and relationships that are related to the spread and mortality rate of COVID-19 around the world. This review should ensure that these innovations and technologies are instantly available to assist decision-making and robust scientific research that will improve the population health diseases outcomes around the globe.

## 1. Introduction

The coronavirus disease (COVID-19) is an infectious disease caused by severe acute respiratory syndrome coronavirus 2 (SARS-CoV-2) [1]. Since COVID-19 emerged in Wuhan, China in December 2019, it has rapidly spread worldwide. It was given pandemic status on 11 March 2020 [2]. While the first informed case was in China, making it the epicenter of the pandemic, COVID has altered its transmission patterns and mutated many times since then [3,4,5].

Lately, India, Europe, and the United States have been reporting the highest number of COVID cases with fast growth in the number of fatalities and confirmed cases [3,4,5]. By 1 April 2020, COVID-19 had infected more than eight hundred thousand (800,000) individuals and caused over forty thousand (40,000) deaths in more than 205 countries and territories [2]; more recently, it has affected more than 614 million individuals and taken more than 6.5 million lives worldwide [6]. COVID-19 has deeply affected the United States, China, and Europe [2].

In addition, the COVID-19 pandemic has effected major socio-economic and health disruptions all over the world [7]. The persistence and emergence of COVID-19 in the United States after the death of more than half a million Americans has undoubtedly affected and altered American life [8]. The coronavirus pandemic has also caused enormous economic, public health, and social damages [9].

We should draw our attention to the fact that the risk factors of COVID-19 are still under investigation, but some demographic factors, such as age, gender, sex, race, marital status, and religion, may play a vital role in increasing both the testing rates and positivity rates within a population. Additionally, the announcement of COVID-19 as a pandemic and the lockdown that occurred at many different levels, from city, to state, to country levels, had a massive influence on our surrounding environment.

It was different than what we usually see and examine [3,10,11]. Despite all the studies analyzing the impacts and effects of this ongoing pandemic, there are fewer that assess the impact of imposing lockdowns and the availability of health and medical data using different geospatial, satellite imaging, and remote sensing assessment techniques [3,10,12].

However, the application and use of remote sensing, satellite imaging, and geospatial analysis techniques and tools platforms provide specialists, practitioners, and the scientific community a large variety of benefits [13]. These benefits can include, but are not limited to, real-time tracking of reported and confirmed case numbers and more understandable and straightforward visualization [13]. In addition, these benefits include spread direction and contact tracing, which can detect the hotspots to control the community spread and dispersion [14,15].

It is worth mentioning that the application of geospatial analysis in public-health-associated problems is not something that was newly presented during the COVID-19 pandemic [16]. Geospatial analysis and geographic information systems (GIS) were used by many different studies in the past. They were even used before the GIS software was born in the mid-1960s [15]. The use of geospatial analytics and GIS include, but is not limited to, visualizing, mapping, analyzing, and detecting patterns of different diseases, especially infectious diseases mapping [15].

Remote sensing is a method that is commonly used to collect physical data to be integrated into a GIS. It is the process of detecting and monitoring the physical characteristics of a specific geographic area by measuring its reflected and emitted radiation at a distance using a satellite or aircraft [17]. Satellite imagery is used to measure, identify, and track human activity [18]. Geospatial is related to the data that are directly linked to specific geographic locations. Moreover, geospatial is a broad term that includes various types of geographic imagery and mapping technology, and GIS is one form of such technology [19].

Satellite imaging is a type of remote sensing that focuses on scientific image capture from space using reflective and emitted electromagnetic (EM) spectrum energy. Remote sensing is a broader category of spaceborne sensing that includes satellite imaging, as well as sensing of non-EM characteristics, such as sea surface temperature, vegetation coverage, cloud, aerosol properties, and other physical attributes. It is important to mention that research groups often characterize their work as one or the other, and so we have included both in the systematic review for completeness [20,21].

Several geospatial, satellite imaging, and remote sensing techniques, software, and methods have been applied to stop the transmission of COVID-19 by contact tracing and imposing lockdowns [16]. John Hopkins University is one of the best examples of geospatial and GIS application during COVID-19 pandemic [13,16,22]. Additionally, local and regional bodies and the World Health Organization (WHO) followed the same direction [16,22].

It is important to mention that in the beginning, using and/or implementing geospatial and GIS techniques was more concentrated on contact tracing and visualizing [3,18]. As more data started to become available, spatial analysis later moved to focus more on economic and environmental aspects, incorporating social perspectives and more advanced analytical tools and techniques [3,18,20].

Additionally, systematic reviews provide a replicable, structured, organized, and systematically integrated landscape proof that could advise practice and policymaking. During the COVID-19 pandemic, one of the major challenges was the scarcity of evidence for public policymaking. These challenges demand a very careful evaluation of the literature on remote sensing, satellite imaging, and/or geospatial analysis in relation to COVID-19 severity.

It is worth noting that systematically assessed evidence is very crucial for the innovation of science as additional research synthesis and primary studies could be learned by the analysis and findings of a systematic review. This knowledge gap has been recognized and intellectualized in this systematic review to improve science, advanced technology, and practice related to a varied range of geospatial procedures, remote sensing, and satellite imaging tools, and methods that are being used in the COVID-19 pandemic.

The purpose of this study is to review the current literature on the application and role of remote sensing, satellite imaging, and/or geospatial analysis for understanding COVID-19 population severity.

## 2. Materials and Methods

### 2.1. Protocol Followed

The review protocol for this systematic review was not recorded in advance of publication.

### 2.2. Final Searched Date

The final search was performed in Scopus and PubMed on Monday 14 March 2022.

### 2.3. Data Sources and Searches

The Preferred Reporting Items for Systematic Reviews and Meta-Analysis (PRISMA) references were followed in directing this systematic review [14]. The keyword search included all types of articles from any year in Scopus and PubMed. It only included articles that are written in the English language and all varieties of studies from any year. Table 1 shows all the search techniques and strategies we used in this systematic review.

### 2.4. Study Selection

The authors carefully and individually screened and reviewed titles and abstracts for eligibility. Both observational and experimental studies were included in the review that were considered eligible during the screening process.

### 2.5. Inclusion and Exclusion Criteria

The titles and abstracts of the identified citations and identified eligible articles were first evaluated by the researchers based on the following criteria. The inclusion and eligibility criteria included the following: COVID-19 is the outcome or the dependent variable of the study, satellite imaging data, remote sensing data, and geospatial data as shown in Table 2. In addition, the exclusion criteria included the following: any paper not written in English, data not related to satellite, remote, or geospatial, and study data that are not related to COVID-19 as shown in Table 2.

We decided to exclude all the articles that discuss the use of remote sensing and geospatial data in COVID-19 and its effect on air quality, air pollution, and environment, since they are out of the scope of our paper. The purpose of this study is to review the current literature on the application of the remote sensing, satellite imaging, and geospatial applications in COVID-19-related research studies.

So, we are not considering studies where COVID has an effect on other things (e.g., effect of COVID on low pollution, high pollution, environment, and air quality). Instead, the studies considered aspects, such as high pollution or other factors, using remote sensing and geospatial analytics, in terms of their effects on COVID severity.

We defined the eligibility of the research studies for inclusion in this systematic review based on screening of the articles. For some articles found in Scopus, we obtained some additional studies by searching their bibliographies and references. Table 2 shows a complete step-by-step breakdown process of the inclusion and the exclusion criteria.

**Table 2 ijerph-20-04298-t002:** Complete breakdown of inclusion and exclusion criteria.

Criteria	Inclusion	Exclusion
Time period	Any period	Not applicable
Language of article	English language only	Any other language
Geographic location/region	Any region	Not applicable
Disease addressed/targeted	COVID-19	Any other disease
Study design	Observational studies and self-report studies	Note, letter, systematic reviews, short survey, book chapter, conference paper, conference review, editorials, erratum, and others
Source of data	Remote sensing, satellite imaging, and geospatial data	Any other source of data
Study outcome/dependent variable	COVID-19	Anything else

### 2.6. Data Extraction and Quality Assessment

The six authors extracted data from the nine included studies. The authors followed the process of selection which was performed in two main steps. First, we read and reviewed the titles and abstracts of the citations by the search query to screen the articles based on the inclusion/exclusion criteria mentioned above. Second, we read and carefully reviewed the full text of the citations selected by the first step, and based on that we decided if the paper was eligible for inclusion or not.

The search criteria did not restrict by publication date; due to the recency of the field, the earliest eligible article was published in 2020. In addition, the researchers collected and extracted the following information from each article that was eligible: author and year, study methods, geographic location, source of data, type of data, major findings, study limitations, and results identified by the authors of the studies. Each article was additionally assessed for its contributions to current practice.

## 3. Results

### 3.1. Data

The initial search of the online databases identified a total of 652 publications (PubMed (n = 55) and Scopus (n = 597)) from the inception of the database to March 2022. A total of 52 duplicates were removed, and an additional 253 records were removed due to non-relevance, as letters, reviews, short surveys, book chapters, conference papers, conference reviews, and editorials were not part of the scope of this study.

An additional 79 records were retrieved after screening references. A total of 347 studies were screened for eligibility based on the title and abstract contents, of which only 17 were sought for retrieval for full analysis. Seventeen full-text articles were assessed for eligibility with nine articles meeting the inclusion criteria for the study as shown in Figure 1.

Figure 1 shows a total of 652 research studies and articles identified through the search conducted in PubMed (55 articles) and Scopus (597 articles). We first removed 52 duplicate articles and 253 articles which were defined as notes, letters, reviews, short surveys, book chapters, conference papers, conference reviews, editorials, and erratum from Scopus. Then, we screened the title and the abstract of the 347 (55 articles from PubMed and 292 articles from Scopus) articles left for eligibility.

A total of 330 articles did not meet the eligibility criteria and were excluded based on the title and abstract review. The 17 remaining articles were assessed in full text for eligibility and data extraction. After a full-text review, seven articles were excluded for the following reasons: two articles were systematic review studies, one article had no health outcome, one article had air quality as the main target of the study (not COVID-19), and four articles did not use geospatial, satellite imaging, and/or remote sensing data in their research study. Finally, a total of nine selected articles were confirmed for data extraction.

**Figure 1 ijerph-20-04298-f001:**
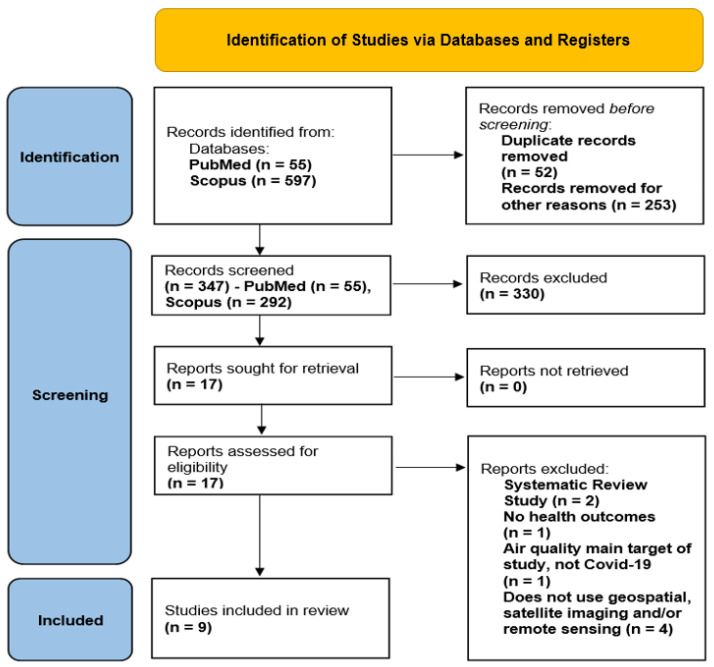
PRISMA diagram shows that a total of 652 articles were identified through the search conducted in PubMed (55 articles) and Scopus (597 articles). It also shows the step-by-step process which lead to the final 9 articles that we selected and confirmed for data extraction.

### 3.2. Study Characteristics

Articles included studies from Europe, Somalia, the USA, Indonesia, Iran, Ecuador, China, and India. The source of data that has been used for these studies mostly relied on governmental health sources and other studies that used geospatial tools to collect the data. Two studies used only satellite imagery data and four studies used remote sensing, while three studies used a combination of both satellite images and remote sensing. Lastly, two studies used spatiotemporal data as the type of data collection. The study methods were wide and varied.

Many studies used reports from healthcare facilities and health records to collate the type of data. For example, Joint Research Center (JRC) as in Amoroso N. et al., Google Earth, Google Maps, and OpenStreetMaps as in Warsame A. et al., CHIRPS USC Santa Barbara and ASTER Imagery as in Purwantara S. et al., and Health: USA FACTS-Delineasi ASTER GDEM 30 m Spatial-MODIS TERRA satellite system as in Johnson D.P et al. In addition, one of the research studies used the National Oceanic and Atmospheric Administration (NOAA) database. Three studies used satellite imagery to collect data.

This systematic review shows that there are significant statistical associations between air pollution (NO${}_{2}$) and COVID-19 mortality [16]; burial rates increased during the pandemic [23]; COVID-19 cases and deaths vary through time [23,24]; there are variations in landscape and meteorological parameters in Indonesia which do not have a significant impact on the spread of COVID-19 [24,25].

Additionally, there is a direct connection between PM2.5 and COVID-19 infections and mortality [25,26]; vulnerable populations that live westwards of active volcanoes in Ecuador have a higher chance of mortality and infection due to the distribution of volcanic ash and spread of COVID-19 [26,27], and the use of satellite remote sensing and GIS can provide practical solutions at an administrative level.

### 3.3. Methods Used in the Articles

Among the nine studies shown in Table 3, Table 4 and Table 5 that we selected, three papers applied a type of machine learning such as support vector machine (SVM), random forests (RF), simple linear regression (LR) model, and multi-layer perceptron (MLP), while six papers employed statistical tools and analysis such as descriptive statistics and Pearson correlation coefficient. Moreover, these different studies were carried out on diverse geographical locations around the word and most of them are country-level data.

It is worth noting that there are so many excellent papers that were excluded, such as the ones mentioned in our inclusion/exclusion criteria Table 2. While the authors were reviewing the articles, they decided that one of the main eligibility criteria was that COVID-19 is the outcome or the dependent variable of the study. We eliminated studies where COVID-19 was the causal/independent variable. Additionally, the study must use satellite imaging, remote sensing, and/or geospatial data.

It is important to mention that COVID severity tied to geospatial-, SI-, or RS-derived independent variables are the key in this paper. Additionally, we excluded a huge body of work on the effect of COVID-19 on air quality, pollution, and environment, as they were the outcome of the excluded studies and not COVID-19.

**Table 3 ijerph-20-04298-t003:** Synthesis of the systematic review results-part A.

**Article Title:** Satellite data and machine learning reveal a significant correlation between NO${}_{2}$ and COVID-19 mortality			**Study Methods:** Random forests (RF), multi-layer perceptron (MLP), support vector machine (SVM), and a simple linear regression (LR) model	
**Source of Data:** Health: Joint Research Centre (JRC) (https://github.com/ec-jrc/COVID-19), accessed on 14 October 2022			**Major Findings:** Significant statistical association between air pollution (NO${}_{2}$) and COVID-19 mortality	
**Author**	**Year**	**Geographic Location**	**Type of Data**	**Study Limitations**
Amoroso N. et al. [24]	2022	European countries—continental scale	Satellite “Sentinel-5p L3”, S-5p data, and climatic data were collected from ERA5	Omitted—variable bias; multicollinearity
**Article Title:** Excess mortality during the COVID-19 pandemic: a geospatial and statistical analysis in Mogadishu, Somalia			**Study Methods:** Imputed missing grave counts using surface area data	
**Source of Data:** Data and satellite imagery (Google Earth, 2021, Google Maps, 2021, OpenStreetMap, 2021)			**Major Findings:** Burial rates increased during the pandemic	
**Author**	**Year**	**Geographic Location**	**Type of Data**	**Study Limitations**
Warsame A. et al. [25]	2021	Mogadishu, Somalia	Satellite imagery, remote sensing, and geospatial analysis	Systematic or random error may have arisen in our method for imputing missing grave counts
**Article Title:** The relationship between landscape and meteorological parameters on COVID-19 risk in a small-complex region of Yogyakarta, Indonesia			**Study Methods:** Simple linear regression and geographic information system (GIS) analysis	
**Source of Data:** CHIRPS USC Santa Barbara and ASTER Imagery			**Major Findings:** Variation in landscape and meteorological parameters in the Yogyakarta area does not have a significant impact on the spread of COVID-19. Ease of mobility in a medium-wide area can encourage the spread more than terrain and climate	
**Author**	**Year**	**Geographic Location**	**Type of Data**	**Study Limitations**
Purwantara S. et al. [27]	2021	Yogyakarta, Indonesia	Remote sensing images	Not Reported
**Article Title:** Analyzing relationships between air pollutants and COVID-19 cases during lockdowns in Iran using Sentinel-5 data			**Study Methods:** Employing SPSS (version 18), descriptive statistics and Pearson correlation coefficient	
**Source of Data:** National Oceanic and Atmospheric Administration (NOAA) database			**Major Findings:** Obvious connection between PM2.5 and COVID-19 infection cases (r = 0.63) and mortality (r = 0.41). Correlation between CO and daily mortality cases (*p* = 0.112) and O${}_{3}$ with the number of infected people was statistically insignificant (*p* = 0.482)	
**Author**	**Year**	**Geographic Location**	**Type of Data**	**Study Limitations**
Rad A.K. et al. [4]	2021	Iran	Remote sensing by employing Sentinel-5P satellite data	Not Reported

**Table 4 ijerph-20-04298-t004:** Synthesis of the systematic review results—continued part B.

**Article Title:** Spatiotemporal Associations Between Social Vulnerability, Environmental Measurements, and COVID-19 in the Conterminous United States			**Study Methods:** Bayesian hierarchical spatiotemporal model, ecological regression, model selection criteria	
**Source of Data:** Health: USA FACTS - Delineasi ASTER GDEM 30 m Spatial - MODIS TERRA satellite system			**Major Findings:** Spatiotemporal character of the pandemic in the US after accounting for specific contributors to social vulnerability, environmental measurements, and spatial and temporal random effects. COVID-19 cases and deaths vary considerably through time and space	
**Author**	**Year**	**Geographic Location**	**Type of Data**	**Study Limitations**
Johnson D.P et al. [26]	2021	County-level in the United States	Remote sensing, landform characteristic, North American Land Data Assimilation System (NLDAS)	Lack of greater temporal resolution with regard to the SVI. Number of zeros the data set for COVID-19 fatalities contains, at least in the initial months
**Article Title:** Volcanic ash as a precursor for SARS-CoV-2 infection among susceptible populations in Ecuador: A satellite imaging and excess mortality-based analysis			**Study Methods:** Statistical comparison of satellite imaging and excess-mortality-based analysis	
**Source of Data:** Not Reported			**Major Findings:** Based on the evaluation of the geospatial distribution of the volcanic ash, the continuous spreading wave of the virus and its mutants will most likely affect vulnerable persons westwards of the active volcanoes	
**Author**	**Year**	**Geographic Location**	**Type of Data**	**Study Limitations**
Toulkeridis T. et al. [10]	2021	Ecuador	Satellite images	Not Reported
**Article Title:** Spatiotemporal spread pattern of the COVID-19 cases in China			**Study Methods:** Local Moran’s I Statistic to delineate the spatial distribution of the weekly new and total cases. Using a generalized additive model (GAM), we linked each potential factor with the COVID-19 cases to quantify its effect on the pandemic	
**Source of Data:** Reported by the National Health Commission of China and the Health Commissions of local governments from 17 January to 20 March 2020. The base map of China was provided by the Resource and Environment Data Cloud Platform (www.resdc.cn, accessed on 3 November 2022). All the spatial data were projected using the Albers equal-area conic projection			**Major Findings:** All results and findings provide valuable insights into the transmission evolution and curbing the spread of COVID-19	
**Author**	**Year**	**Geographic Location**	**Type of Data**	**Study Limitations**
Feng Y. et al. [5]	2020	China	Spatiotemporal data	Not Reported

**Table 5 ijerph-20-04298-t005:** Synthesis of the systematic review results—continued part C.

**Article Title:** A new method for identifying and mapping areas vulnerable to COVID-19 in an armed conflict zone: Case study north-west Syria			**Study Methods:** The Risk of Vulnerability to COVID-19 in War Zones Index. This index was calculated based on factors using spatial data	
**Source of Data:** Data on healthcare facilities and their status in north-western Syria were extracted from Syria’s health sector bulletin, developed by the World Health Organization, together with the functional status of each facility. Data on the bombings that took place in this zone over the previous six months were extracted from the Armed Conflict Location and Event Data Project (ACLED) website, as was the fatality rate of each attack			**Major Findings:** This paper presents a new method for identifying and mapping vulnerable areas. This method can be used in any conflict zone in the world. The map resulting from this paper can be used to manage the pandemic in this region by preparing the most vulnerable zones with the necessary health facilities and protective measures	
**Author**	**Year**	**Geographic Location**	**Type of Data**	**Study Limitations**
Mobaied S. et al. [11]	2020	Northwest Syria	Remote sensing and spatial modeling	Not Reported
**Article Title:** Reporting the management of COVID-19 threat in India using remote sensing and GIS-based approach			**Study Methods:** GIS proximity analysis and network analysis	
**Source of Data:** Reports from GIS agencies of the state			**Major Findings:** Satellite remote sensing and GIS can provide practical solutions at an administering level	
**Author**	**Year**	**Geographic Location**	**Type of Data**	**Study Limitations**
Kanga S. et al. [12]	2020	Ramganj, Jaipur, India	Using an integrated satellite remote sensing, geographic information system (GIS), and local-knowledge-based approach	Not Reported

### 3.4. Geographic Location and Type of Data

One study was performed on a continental scale of European countries, three studies were carried out in Africa, including the one in Mogadishu, Somalia. Additionally, one study was performed on a country level in the United States, three studies were performed in Asia at the following locations: Yogyakarta, Indonesia; China; and Ramganj, Jaipur, India. Moreover, two studies were carried out in two different countries in the Middle East, in Iran and Syria. Finally, one study was performed in South America, in the country of Ecuador.

In summary, two studies were carried out in Europe and Ecuador using satellite imaging, three studies in the United States, Indonesia, and Syria using remote sensing, one study in China using spatiotemporal data, and three studies in India, Iran, and Somalia using both satellite imaging and remote sensing.

Figure 2 shows the world map with the geographic location by the type and source of data. The red color represents the satellite imaging data type that was used in Europe and Ecuador. The green color represents the remote sensing data type which was collected in the United States, Indonesia, and Syria.

The yellow color represents the spatiotemporal kind of data that were used in China [5]. Spatiotemporal data are the data related to both space and time. In this study, the authors used spatiotemporal data to explore the spread pattern of COVID-19 and spatial clustering in China and analyze the relationships between the number of cases and its potential influencing factors [5].

**Figure 2 ijerph-20-04298-f002:**
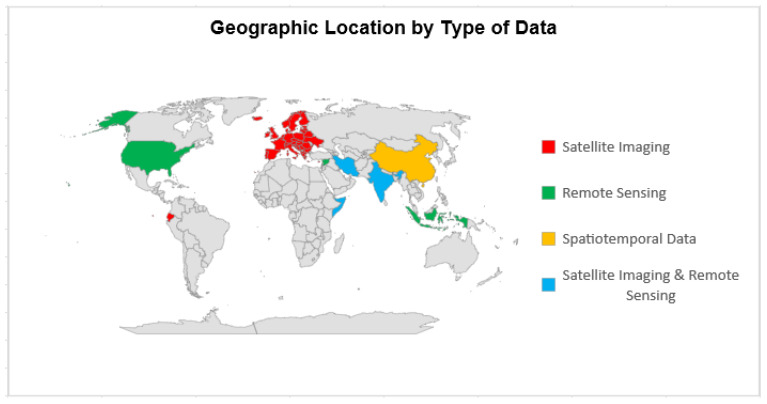
World map diagram shows the geographic location for each study by the type/source of data. Two studies in Europe and Ecuador used satellite imaging, three studies in the United States, Indonesia, and Syria used remote sensing, one study in China used spatiotemporal data, and three studies in India, Iran, and Somalia used both satellite imaging and remote sensing.

The spatiotemporal data in this study include the number of COVID-19 cases and their locations to help identify the spatial spread of the pandemic [5]. The daily number of new and total COVID-19 cases was collected for each prefecture-level and county-level city in China from the National Health Commission of China [5]. Lastly, the blue color represents the studies that used both satellite imaging and remote sensing in conducting their data analysis in Iran, India, and Somalia.

## 4. Discussion

Our research study does not only provide an overview and analysis of how remote sensing, satellite imaging, and geospatial data have been used, but also provides pointers and draws scientists’ and researchers’ attention to how remote sensing, satellite imaging, and geospatial analysis could be more efficiently and actively used in COVID-19 research and other population health diseases in the future.

It is important to note that the use and application of remote sensing, satellite imaging, and geospatial analysis has significantly influenced the understanding of COVID-19 for policymakers, the scientific community, as well as the public in building a long-term response to the ongoing pandemic [15,16].

We reviewed how remote sensing, satellite imaging, and geospatial analysis tools and techniques were used previously in COVID-19 research studies. We found that the majority of those studies used remote sensing only, satellite imaging only, or both remote sensing and satellite imaging or spatiotemporal data to visualize the geospatial distribution and the pattern of COVID-19 spread. Some studies also used a significant statistical analysis to define the association between air pollution (NO${}_{2}$) and COVID-19 mortality [16,23].

Other studies used remote sensing, spatial, and temporal random effects to show that COVID-19 cases and deaths vary considerably through time and space [23,24]. Moreover, there were several studies that focused on different models to simulate and predict many aspects of COVID-19 using remote sensing, satellite imaging, and/or geospatial techniques.

Our findings support the vital role of accurate reporting and availability of data during a global pandemic for geospatial analysis which enables real-time decision-making for preventing public health crises. Therefore, we found that in developed and wealthy countries, such as the United States and Europe, studies covered larger geographic areas where COVID-19 data are mostly publicly available.

Studies covered multiple states and countries and were performed on a continental scale as shown in Amoroso N. et al. [24]. On the other hand, in developing or less-developed and poor countries, such as Syria, Somalia, and Iran, studies covered smaller geographic regions such as single cities due to the restricted accessibility, limited availability of the data (data unavailability), cost, limited research budget, and technological issues.

Another major challenge of regional and global studies is the under-reported number of COVID-19 confirmed cases and deaths, especially in underdeveloped, poor, and low-income geographic regions. This could be due to the low detection coverage of COVID-19 which might lead to skewness in the results.

The results from the reviewed articles demonstrated that those regional and global studies cannot integrate the controlling measures enforced by many different governments. Those measures have significant impacts on the spread of COVID-19. It is worth noting that a substantial limitation for prediction- and modeling-focused research are the low testing issues and the inconsiderate government control measures which might create a biased result. Future studies should consider the role of government policies and control measures in their modeling.

Deep learning, convolution neural networks (CNNs), deep convolution neural networks (DCNNs), and data mining research techniques would be useful in forecasting and analyzing the geospatial pattern, satellite imaging, and/or remote sensing of a population’s COVID-19 severity [28]. It is worth mentioning that deep learning models are more difficult to build compared to traditional machine learning models, since they use complex multilayered neural networks [29].

Furthermore, deep learning methods are more advanced models and they require more data richness. Additionally, some of those methods have the ability to concurrently learn feature engineering while learning their classification or regressor capability. These characteristics will increase the use of deep learning methods in the geospatial and population health diseases future research studies [30,31].

Additionally, we believe that the use of remote sensing, satellite imaging, and geospatial data analysis may improve other practical measures and increase the scope of scientific exploration on many research topics. The evidence shown in this systematic review highlights the application and use of remote sensing, satellite imaging, and geospatial analysis techniques and tools for addressing research questions related to the COVID-19 pandemic.

However, there is little evidence on how remote sensing, satellite imaging, and geospatial analysis can be used to target populations and for delivering digital interventions for individuals. These technologies, developments, and innovations might take a longer time to appear, but precision sciences and medicine and their tools and applications may bring those innovations and technologies closer to everyday practice.

Moreover, several data sources and methods are used in different research studies to collect, geocode, and then analyze the data which provide useful and meaningful intuitions on how geospatial data could be applied and harmonized in addressing population-health-diseases-based problems. Geospatial data can help epidemiologists to respond to disease outbreaks faster and to better understand community health problems and disease diffusion, as well as improve leadership awareness to public health issues.

For example, the government of India during the pandemic lockdown used geospatial data to map COVID-19-affected zones and prepare hotspots maps of the pandemic along with utilities maps to manage the humanitarian crisis within such zones effectively [12]. Additionally, in the country of Ecuador, the researchers used satellite imaging and excess-mortality-based analysis to conduct the very first study of its kind combining the social and spatial distribution of COVID-19 infections and volcanic ash distribution [10].

Additionally, the Johnson D.P et al. study results show how researchers successfully fitted a Bayesian hierarchical spatiotemporal model to COVID-19 cases and deaths at the county level in the United States [26]. Another research study was conducted in Yogyakarta, Indonesia by several agencies using the primary and secondary data obtained from remote sensing images [27]. This research study by Purwantara S. et al. shows that there is no significant impact between the spread of COVID-19 and the variation in meteorological parameters and landscape in the Yogyakarta area in Indonesia [27].

Furthermore, integrating remote sensing, satellite imaging, spatiotemporal, and geospatial analysis in COVID-19 research may facilitate real-time decision-making to utilize resources and prevent public health crises whenever required. A very important lesson that could be learned from the existing studies is to develop local and global disaster awareness and preparation plans.

For example, during the pandemic, Mobaied S. et al. used remote sensing and spatial modeling for identifying and mapping vulnerable areas in an armed conflict zone to limit the risk and predict the spread of COVID-19 in north-west Syria [11]. Additionally, the results and implications of Toulkeridis T. et al.’s study in Ecuador will play a vital role in assisting the countries to identify aforementioned vulnerable parts of society, especially if the given volcanic settings and geodynamics are similar [10].

The Amoroso N. et al. research article used satellite imaging data to conduct its study. The study results disclose a significant statistical association between COVID-19 mortality and NO${}_{2}$ air pollution [24]. It also reveals the vital role played by socio-demographic features such as the number of hospital beds, gross domestic product per capita, and the number of nurses [24].

In addition, the results of the study conducted in Iran by Rad A.K. et al. clearly show how governments should handle and control the outbreak of COVID-19 in several ways. Those ways could include, but are not limited to, environmental conservation strategies as well as implementing efficient quarantines [4]. Moreover, the findings in Feng Y. et al. provide valuable and useful insights to better understand the several changes in COVID-19 transmission. It also shows the appropriate actions and precautions that could be taken to control the spread of the COVID-19 pandemic around the world [5].

This might allow practitioners, health care specialists, and policymakers to leverage and support the use of remote sensing, satellite imaging, and geospatial-based advanced data analytics for modeling large-scale public health emergencies [3,24]. Such innovations, tools, and techniques may require strengthening technological capacities and abilities, especially in low- and middle-income geographic regions and countries. Those countries share the same local and global problems due to limited resources, lack of funding, and scarcity of data [3,24].

Lastly, it is vital to mention that the research field where those research methods are used to define features related to population health diseases such as COVID-19 is somewhat new, and the number of study publications in this particular research area is small [3,25]. Furthermore, most of the studies are very limited in geographic scale, where seven out of the nine articles used one or two particular cities to conduct their study analysis [3,25].

However, two article studies used a larger geographic scale (USA and Europe), where the potential exists for large geography analysis using these tools [3,25]. Additionally, these studies are being conducted around the world in countries that are rich, wealthy, and developed, as well as in countries that are developing (less developed), poor, and unwealthy. Furthermore, studies have been conducted in geographic regions under peace and in other regions where people are facing civil conflict [3,11]. This shows great potential for applying RS, SI, and geospatial analysis for health matters in most under-served populations [11,25].

We should note that this systematic review has various limitations. Those limitations should be tackled and addressed in future research work. First of all, we did not cover the preprints that did not undergo peer review, but we focused on peer-reviewed publications [3,11]. Since those research studies were also excluded from this systematic review, this might provide further insights into the evidence landscape.

Second, the selection of databases and keywords could have excluded some studies that were indexed in other databases or used non-specified keywords, which were beyond the scope of this review [3,12]. Third, and lastly, we did not take the risk of bias between and within different studies into consideration [3,12]. We recommend that future data syntheses on geospatial, remote sensing, and satellite imaging topics should consider the risk of bias in the scientific literature around those topics.

## 5. Conclusions

This systematic review assessed the current literature on the use and application of remote sensing, satellite imaging, and geospatial analysis in the research related to the COVID-19 pandemic. We are confident that the body of literature on the use of remote sensing in COVID-19 epidemiology in terms of severity and mortality is still new, very small, unique, and novel. Additionally, it will help with exploring the scope of integrating these technologies and techniques in the current research studies as well as future studies and practice.

In the era of advanced technology and the rapid digital revolution, it is very crucial to exchange technological innovations, advancements, and developments across all scientific disciplines. The application of remote sensing, satellite imaging, and geospatial tools and technologies during the COVID-19 pandemic provides knowledgeable assessments of how global and societal problems can be understood using the techniques.

Moreover, these techniques require re-evaluating the current strengths and weaknesses of the used tools and applications. It is extremely important to strengthen the institutional abilities to influence remote sensing, satellite imaging, and geospatial innovations and technologies in order to improve the research and development of defining features that could be related to population health diseases.

This will encourage the research communities to collaborate and work together to face the challenges of the ongoing COVID-19 pandemic. Lastly, future technological advancements and innovations should be built on the lessons learned during the COVID-19 pandemic. In addition, those research communities should make sure that these innovations and technologies are instantly available to assist decision-making and robust scientific research that will improve the population health disease outcomes around the globe.

## Figures and Tables

**Table 1 ijerph-20-04298-t001:** Search strategies and techniques.

Database	Search Term	Resulted Articles	Articles Selected	Comments
PubMed	((Satellite Imaging[Title/Abstract]) OR (Remote Sensing[Title/Abstract]) AND (Covid-19[Title/Abstract]))	55	55	(All relevant documents were duplicates from Scopus search)
Scopus	TITLE-ABS-KEY (((“satellite imaging”) OR (“remote sensing”)) AND (“COVID-19”))	597	292	Number of duplicate records: 52. Not articles: 253 (note, letter, review, short survey, book chapter, conference paper, conference review, editorial)

## Data Availability

Data from this study will be made available upon request.

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
