# Peer review of "The Role of Remote Sensing and Geospatial Analysis for Understanding COVID-19 Population Severity: A Systematic Review"

_ijerph, 2023, doi:10.3390/ijerph20054298_

Round 1

Reviewer 1 Report (New Reviewer)

1. The author suggests that these early papers focus on contact tracing and visualization with remote sensing images and geoinformation in the beginning from line 78 to 82.  Please supplement the changes of research perspectives in later period.

2. these removed 253 records  should be divided into groups and shown the statistic data  of each group in fig. 1, and the excluded 330 articles should be processed in the same way, which is import to judge whether the literature selection method is scientific and rational.

3. For the contents of 172-184 lines, it is recommended to organize it into a table for easier analysis and reading.

4. in figure 2, the term spatiotemporal data is used ,but the author doesn't explain exactly what data is included. Does it contain satellite images or other geospatial information? Additionally, it is generally believed  that satellite images are one data source obtained by remote sensing, how to  distinguish between these two concepts shown in figure 2?

5. The contents of Table 3, 4, and 5  lack adequate explanation and analysis. For example, why divide results into categories A,B, and C?

6. It is suggested to sort out the discussion part, like which are the viewpoints in the 9 articles and which are the viewpoints in other papers?

Author Response

Reviewer 2 Report (New Reviewer)

The document is well structured, it is coherent, the title, aim and methodology, the which supports the results, discussion and conclusion. 

Comments:

There are some paragraphs in the discussion that have no references.

In the document there are very long paragraphs, they should be shortened to a maximum of 6 lines per paragraph

In the discussion section, the last paragraphs are not supported by bibliographical references, one might think that they are only contributions from the actors and there is no discussion.

Author Response

This manuscript is a resubmission of an earlier submission. The following is a list of the peer review reports and author responses from that submission.

Round 1

Reviewer 1 Report

What is the difference between Remote sensing and Satellite imaging? Isn't it basically about the same content?

Among hundreds of existing studies cited or non-cited, nine papers are selected in Table of Synthesis of the Systematic Review Results. However, the criteria selected for analysis are not clear.

Compared to the research title, the research direction or the applied method deals with small content.

Although this paper has been a meta-analysis, it is not easy to find practical research novelty.

Reviewer 2 Report

The article, despite its direct title, does not contain a substantive review of the literature. The basic error in the methodology is the use of a simple keyword search that does not take into account the entire text of the paper.

For example, the work entitled "COVID-19 pandemic and environmental pollution: A blessing in disguise?" has not been included in the literature review - and it is a work cited more than 899 times.

I suggest the authors of the work use a different method of searching for sources.

The topic undertaken by the authors is important, but the current result of the works found on covid + remote sensing is very low and unjustified by the facts.

The work will be suitable for publication after a significant extension of the literature review. An example can be a work on a similar topic: "The role of remote sensing during a global disaster: COVID-19 pandemic as case study"